# Enhancing Urban Landscape Design: A GAN-Based Approach for Rapid Color Rendering of Park Sketches



**Ran Chen** [1], **Jing Zhao** [1,*], **Xueqi Yao** [1], **Yueheng He** [1], **Yuting Li** [1], **Zeke Lian** [2], **Zhengqi Han** [1], **Xingjian Yi** [1] **and Haoran Li** [3]

[1] School of Landscape Architecture, Beijing Forestry University, Beijing 100083, China; chenran705367787@bjfu.edu.cn (R.C.); yxq18210187099@bjfu.edu.cn (X.Y.); candyhyh808@bjfu.edu.cn (Y.H.); liyuting1625571259@bjfu.edu.cn (Y.L.); zenkii@bjfu.edu.cn (Z.H.); yixingjian0906@bjfu.edu.cn (X.Y.)
[2] Ningbo City College of Vocational Technology, Ningbo 315100, China; lianzeke@nbcc.edu.cn
[3] China United Network Communications Group Co., Beijing 100031, China; lihr21@chinaunicom.cn
[*] Correspondence: zhaojing@bjfu.edu.cn; Tel.: +1-3811-993-788

**Abstract:** In urban ecological development, the effective planning and design of living spaces are crucial. Traditional color plan rendering methods, mainly using generative adversarial networks (GANs), rely heavily on edge extraction. This often leads to the loss of important details from hand-drawn drafts, significantly affecting the portrayal of the designer's key concepts. This issue is especially critical in complex park planning. To address this, our study introduces a system based on conditional GANs. This system rapidly converts black-and-white park sketches into comprehensive color designs. We also employ a data augmentation strategy to enhance the quality of the output. The research reveals: (1) Our model efficiently produces designs suitable for industrial applications. (2) The GAN-based data augmentation improves the data volume, leading to enhanced rendering effects. (3) Our unique approach of direct rendering from sketches offers a novel method in urban planning and design. This study aims to enhance the rendering aspect of an intelligent workflow for landscape design. More efficient rendering techniques will reduce the iteration time of early design solutions and promote the iterative speed of designers' thinking, thus improving the speed and efficiency of the whole design process.

**Keywords:** hand-drawn sketch; image color rendering; generative adversarial networks; data augmentation; landscape design

## 1. Introduction

The construction of urban living environments is a crucial aspect of realizing urban ecological civilization. It directly impacts residents' quality of life and the sustainable development of cities. Park green spaces, as essential elements for enhancing the ecological environment and city image, are among the city's most vital resources. In the planning process of urban green spaces, designers integrate various design concepts, such as site space layout, landscape nodes, and road traffic, into hand-drawn plans. These plans directly express the designer's thoughts, and the rendered colorful plans illustrate the design effect. This process aids communication with multiple parties and enhances the design's quality and level. However, the current workflow from the research stage to the color flat drawing stage is often time-consuming. Therefore, improving the efficiency of plan rendering is a significant challenge in the current design process and the primary motivation and goal of this study.

To address this challenge, we explored the usefulness of our study based on traditional landscape design methods and their inherent challenges. In the traditional landscape design process, designers typically go through stages like site analysis, concept development, design evolution, and drawing production to complete a project [1,2]. Initially, designers conceive the spatial arrangement of landscape elements based on the site conditions and

client requirements. They often sketch designs by hand, as it better captures smooth curves. Eventually, these sketches are turned into colored flat plans using computer software to present to clients. Since clients usually lack professional knowledge, designers need to use these colored plans to explain the overall design. The process requires repeating these steps in communication with clients. The step from sketch to colored plan is time-consuming, posing a major challenge. Therefore, our study aims to integrate intelligent tools into the traditional process, enhancing the efficiency of sketch rendering and accelerating the process of design thinking and scheme iteration.

In recent years, generating images with appropriate colors and textures based on sketches has become a research hotspot. The advancement of machine learning has greatly facilitated designers. With the evolution of artificial intelligence, many painting tasks are now accomplished using deep-learning technology. Generative adversarial networks, an innovative architecture, have found widespread application. However, landscape design drawings often struggle to render satisfactory results due to the irregularity of elements, intricate details, and limited available corresponding data. Currently, designers still need to manually complete it through software such as Photoshop, relying on their perception of color matching and the coordination of different detail textures. If deep learning can assist designers in quickly completing the design, it will significantly improve design efficiency and generate more application value. However, the drawing of landscape design diagrams makes it difficult to render better results due to defects such as the weak regularity of elements, complexity of details, and less corresponding data available. Currently, designers are still required to complete the design manually through Photoshop and other software based on the perception of color matching and the coordination of different details and textures. If deep learning can assist designers to quickly complete the design, the design efficiency will be greatly improved and generate more application value.

To address these issues, this study proposes an automatic plan rendering system based on generative adversarial networks, combining the features of machine learning and landscape planning and design. We use hand-drawn line-draft plans and color plan design drawings for training. Compared to the edge extraction commonly used in other studies, hand-drawn line-draft maps are more likely to reflect the actual semantic structure of images, conform to the aesthetics and habits of designers, and facilitate us in screening out the generation algorithm most suitable for the actual design process. After selecting the appropriate algorithm, we built a data enhancement module to optimize the rendering system. The main contributions of this paper are as follows: (1) Based on Pix2pix and CycleGAN, we built a fully automated park plan rendering system and screened out algorithms suitable for the design process. (2) By employing data augmentation models, we were able to expand our small sample size and optimize the training process. (3) We analyzed the differences and similarities in the results generated by various algorithms and sample data volumes. (4) We utilized a novel approach to visualize urban park green spaces, using hand-drawn line drafts as experimental data. (5) Our proposed automated rendering system enhanced the speed and quality of sketch rendering in landscape design. Integrating this system into the design development and drawing production stages of traditional landscape design processes can improve the overall design efficiency.

The remainder of this paper is organized as follows: Section 2 provides a review of the related literature. Section 3 presents our proposed architecture and training methodologies. Section 4 details the experiments conducted and their respective results. Section 5 discusses the related research. Finally, Section 6 elaborates on the results and potential future work.

## 2. Related Work

Our work is primarily related to two research areas: image coloring and conditional generative adversarial networks. This section provides a comprehensive review of the pertinent literature in these fields.

## 2.1. Image Coloring

Image coloring, a technique that infuses grayscale or sketch images with color and texture, holds significant value in the realm of nonphotorealistic rendering [3,4]. However, image coloring presents an ill-posed problem due to the potential for one image to correspond to multiple color schemes. Consequently, conditional information is required to generate vivid and reasonable results. The earliest image coloring methods were grounded in CG methods, employing physical simulation or programming to emulate effects such as watercolor and oil painting, or utilizing feature lines for line rendering. For instance, Chu and Tai leveraged GPU computations to generate watercolor effects [5]. While this method can yield realistic outcomes, it is computationally expensive, complex in design, and challenging to adapt to varying styles and scenes. Traditional methods, such as LazyBrush [6], can only manage simple shape line drafts, and issues like unnatural colors or vacancies arise when dealing with complex patterns. Over the past decade, deep-learning methods have been employed for image coloring tasks. Researchers like Deshpande utilized variational autoencoders (VAEs) to learn the low-dimensional embedding of the color domain [7], thereby constructing a conditional model for generating diversified coloring results. Conversely, researchers like Mouzon integrated variational methods with convolutional neural networks (CNNs) to design a fully automatic image coloring framework [8].

With the advancement of deep learning, researchers have begun to employ generative adversarial networks (GANs) for image coloring across various fields, yielding notable results. In the realm of portrait synthesis, Fang and colleagues concentrated on synthesizing critical identity markers such as eyes and noses [9]. They utilized CycleGAN to generate high-resolution images, transitioning from sketches to facial photos. Simultaneously, in the medical sector, Long and others proposed an enhanced CycleGAN that incorporated perceptual loss [10]. This successfully facilitated the interactive translation of echocardiograms and enabled arbitrary conversions between sketch images and ultrasound images. Subsequently, GAN-based coloring methods were applied to painting tasks. For instance, Peng and others employed the CE-CycleGAN framework to process edge contours and transform landscape photos into Chinese landscape painting styles [11]. Sun and others developed a system capable of fitting edges based on semantic label maps [12], generating exquisite paintings of a specific type. Ren and others used a superpixel segmentation algorithm to optimize the coloring scheme of anime line drafts, addressing issues such as color diffusion and color loss [13]. Wang and others proposed a Thangka color simulation algorithm (SMAC-CGAN), which restored the actual colors of Tibetan paintings based on line drafts, resolving the halo problem [14]. Additionally, GAN technology has been applied to complete the automatic generation process from line sketches to color pictures based on nonspatially corresponding data. For example, Chen and others constructed the SketchyGAN to transform rough sketches of 50 categories, including various animals and daily necessities, into near-real color pictures [15]. Moreover, using similar images as virtual references, Wang and others [16], as well as Lee and others [17], have successfully completed the color transfer of the same type of sketch. This demonstrates the potential of these techniques in various applications.

Existing methods primarily focus on the coloring of grayscale painting images or line drawings, with less emphasis on the rendering of design drawings. In the landscape architecture domain, rendering design sketches involves transforming hand-drawn sketches into images with color, texture, and shadow effects to enhance the expressiveness and appeal of the design. This differs significantly from standard line-drawing coloring, as the lines and colors in design drawings carry landscape element information. These elements not only reflect the designer's creativity and thought process but also the functionality and form of the design. Therefore, rendering landscape design sketches requires a comprehensive consideration of the design's semantics, style, and rules, rather than merely coloring the lines.

In various design fields, researchers utilize different algorithms for rendering design sketches. A common approach is based on generative adversarial networks (GAN), lever-

aging deep learning to automatically learn design features from data and generate realistic images. In architecture, GAN methods have been used for rendering architectural effect and layout drawings. Qian et al. [18] proposed a GAN-based method to render sparse architectural sketches into textured effect drawings. Concurrently, their team also presented a generation process from user-preferred geometric shapes to sketches to high-rise architectural renderings [19]. However, when generating architectural images, the GAN model tended to distort the surface instead of maintaining the original linear structure. To address this issue, Zhao and others combined the Y-shaped GAN and denoising diffusion implicit models [20]. For rendering flat layouts within given boundaries, the system proposed by Wu and others can automatically and efficiently generate residential layout floor plans [21]. Building upon this, Huang Weixin [22] and colleagues used the GAN variant model Pix2pixHD to generate house-type diagrams based on room layouts. On a larger scale, Yang Liu [23] similarly utilized this approach to generate apartment floor plans based on function partitioning diagrams of youth apartments.

In the field of landscape architecture, GAN methods have also been applied to render design line drawings. Zhou et al. [24] used CycleGAN to render different types of plots in parks from color block diagrams to textured color maps, but faced issues with detail recognition, monotonous planar colors, and poor transitions. Our study attempts to compare different algorithms to determine which is more suitable for rendering landscape design drawings. We employed two distinct GAN models, namely, CycleGAN and Pix2pix, chosen for their respective advantages and applicable scenarios. CycleGAN is suitable for situations without paired data, while Pix2pix handles cases with paired data.

Regarding data selection, current image coloring tasks based on deep learning primarily obtain line-sketch datasets through edge extraction. Some researchers use the Canny algorithm to extract edges from real images to complete the coloring task. For instance, Zou and others used traditional image processing technology, Canny edge detection, to extract the line sketches of the Forbidden City patterns [25]. They completed the color restoration of the details of the Forbidden City murals. In terms of design, Sun and others also used the same method to extract the edge contours of billboards and icons for coloring [26]. In painting, because the line sketches obtained by the Canny algorithm cannot meet professional requirements, Aizawa and others used a model specifically based on the extraction of painting image line structure "LineDistiller" to build a dataset [27]. However, the XDoG algorithm used by Golyadkin and Makarov in comic coloring research could generate images with more visual appeal compared to it, which is more suitable for artistic line coloring [28]. Similarly, Zhao and others used this algorithm to obtain edge data for architectural sketch coloring [20]. When Li and others did the rendering of interior effect line drafts [29], some data were extracted by the Canny algorithm, and other data were nonmatching line drafts. Most researchers choose to obtain line-draft sketches by applying algorithms to images for the edge extraction in coloring tasks, and fewer researchers use corresponding hand-drawn sketches with more details. However, in the coloring of plan design drawings, hand-drawn line sketches have richer structures and light and shadow effects than computer-generated edge contour images, which can better express design effects. We choose hand-drawn line-draft sketches as the original data from the actual design process.

## 2.2. Conditional Generative Adversarial Networks

### 2.2.1. GAN Generation

Generative adversarial networks (GANs), proposed by Goodfellow [30] in 2014, represent a neural network framework grounded in game theory. This framework is unique in its inclusion of two competing subnetworks: the generator and the discriminator. The generator's role is to create virtual data that appears real from random noise, while the discriminator is tasked with discerning the authenticity of the input data. During the training process, these two subnetworks operate in an adversarial manner, striving for a Nash

equilibrium. Consequently, the generator can produce virtual data capable of deceiving the discriminator.

### 2.2.2. Pix2pix and CycleGAN

Currently, the application of generative adversarial networks (GANs) in landscape design generation remains somewhat uninterpretable. This paper primarily focuses on Pix2pix and CycleGAN to construct a more scientific and controllable plan rendering design. Pix2pix, through its label set, can control the generation results, making it widely applicable in layout scheme design generation. For instance, contour images have been used as labels to generate robust arrangements for residential areas [31]. Additionally, related design indicators have been introduced to control the generation of urban land distribution schematic diagrams [32]. However, while Pix2pix is a supervised learning model that supports conditional control, it lacks the ability to explain the extraction of hidden features. On the other hand, CycleGAN, an unsupervised learning model, emphasizes adversarial training and cycle consistency loss. It can autonomously extract and explain implicit features. It has been extensively used in image style transfer, image super-resolution, image translation, and other fields. Chen et al. [33] in their research developed a park layout generation system using CycleGAN technology. The system primarily renders color block layouts, but it does not address the rendering of hand-drawn line design drawings, a common aspect in practical engineering design. In rendering color block layouts, the focus is on the overall visual effect of color and spatial element arrangements. However, the rendering of hand-drawn line drawings presents a more complex challenge, requiring the precise expression of line texture and detailed color nuances. Moreover, sketch rendering plays a crucial role in the design process of landscape architects, needing to reflect not just the precision of computer vision but also to aid designers in expressing their creative ideas. In light of these considerations, this study compared the Pix2Pix and CycleGAN algorithms and established an automated hand-drawn line-drawing rendering system based on GANs aimed at better addressing these challenges.

## 3. Methodology

### 3.1. Analytical Framework

Figure 1 presents the comprehensive research framework for the line-sketch rendering design scheme generation system proposed in this paper.

During the dataset construction phase, the quality and quantity of the dataset significantly influence the design learning of the algorithm. Given the challenge of obtaining paired park plan layout and design scheme data, we enhanced the black-and-white line-sketch data using CycleGAN. This enhancement expanded the original 652 pairs of park data to a dataset of 1699 pairs, thereby constructing a more diverse and accurate database for subsequent generation experiments.

In the model construction phase, we employed two neural networks, Pix2pix and CycleGAN, for image generation. The corresponding sketch and design scheme datasets were simultaneously input into the neural network for initial training of the small-sample data rendering system. However, the output results were suboptimal, leading us to add a data enhancement module based on CycleGAN. Consequently, we constructed an optimized plan rendering system using the expanded large-sample data.

During the application phase, the black-and-white line sketch is input into the model. The algorithm then automatically generates a rendering design scheme of the corresponding style within seconds.

**Figure 1.** This is the process of building a line-sketch rendering system.

The following sections will provide a detailed discussion of each step.

### 3.2. Data Collection and Preparation

Due to the difficulty in discerning image details in large-scale design plans, this experiment uses park green space plans as its subject. Since publicly available data on green space design are limited, we collected various types of public training data from design websites and other channels. After initial screening and processing, these were used as our training dataset. The aim of our experiment is to improve the rendering efficiency in landscape design workflows. Therefore, we chose black-and-white hand-drawn sketches, commonly used in design, as our input data instead of colored images. Ultimately, we gathered 152 pairs of black-and-white hand-drawn sketches and design plans as training samples and standardized the dataset into $512 \times 512$ pixel jpg images. Part of our experimental data is shown in Figure 2.

### 3.3. Sketch Rendering Model

Our approach employs Pix2pix and CycleGAN to establish an automated plan rendering system. These are prevalent techniques in style transfer-related research, yet they exhibit substantial differences in their learning methodologies.

#### 3.3.1. Supervised Learning Based on Pix2pix: Working Principle and Process

Pix2pix is developed on the foundation of conditional generative adversarial networks (CGANs). It treats the input image as a condition, learns the mapping relationship between the input and output images, and consequently generates the results [34].

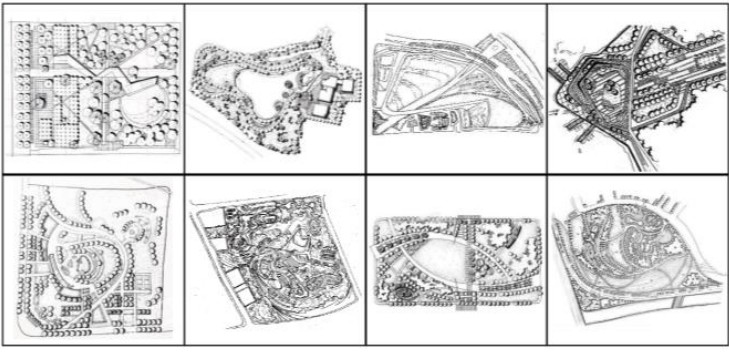

(a)Park sketch dataset

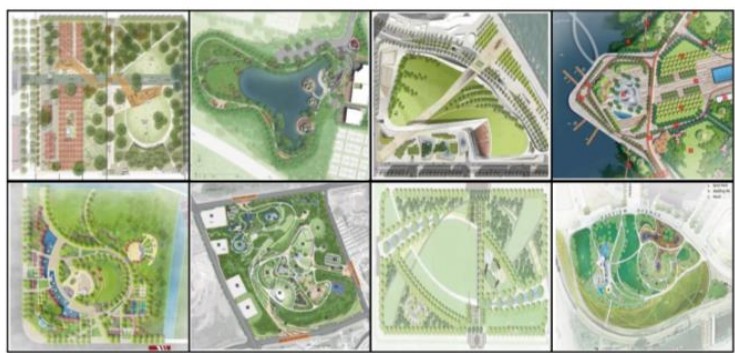

(b)Corresponding design dataset

**Figure 2.** These are the representative examples of hand-drawn sketches and corresponding design drawings dataset: (**a**) park sketch dataset; (**b**) corresponding design dataset.

The fundamental structure of the Pix2pix algorithm, as depicted in Figure 3, comprises two components: the generator (G) and the discriminator (D). The generator employs the U-net framework for feature extraction and upsampling. The discriminator, on the other hand, utilizes PatchGAN to discern whether the local details in the generated scheme closely resemble reality.

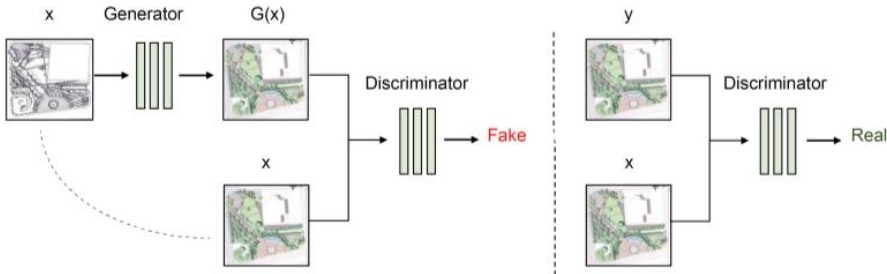

**Figure 3.** This is the workflow of Pix2pix.

3.3.2. Supervised Learning Working Principle and Process Based on CycleGAN

CycleGAN, a type of generative adversarial network (GAN), is utilized for unsupervised image generation. Unlike Pix2pix, CycleGAN does not necessitate paired data input to learn the mapping relationship from one image domain to another [35].

The framework of the CycleGAN algorithm, as illustrated in Figure 4, comprises two generators, Generator A and Generator B, and Discriminator B. The entire process encompasses two discrimination stages: (1) Ascertain the correlation between the result and input through Generator A and Generator B. (2) Compare the discriminator with the real reference scheme to judge whether the generated style is a design scheme or a

hand-drawn line draft. In the first stage, the park line-draft plan is input into Generator A to obtain Result A (real image to fake image). Subsequently, Generator B generates the park line-draft scheme Result B (fake image to fake image) in reverse according to Result A. Through continuous iteration, the calculation error between the original line-draft plan and Result B is reduced, enabling the model to learn the implicit features in the image and generate the most realistic result. In the second stage, Result A is input into Discriminator B. After comparing it with the real design scheme, it is determined whether the output design scheme A is real or fake.

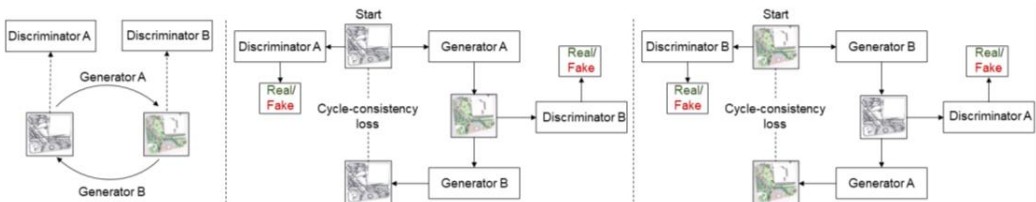

**Figure 4.** This is the workflow of CycleGAN.

*3.4. Training*

The entire training process, as depicted in Figure 5, comprises three stages: pretraining, data augmentation, and optimization training.

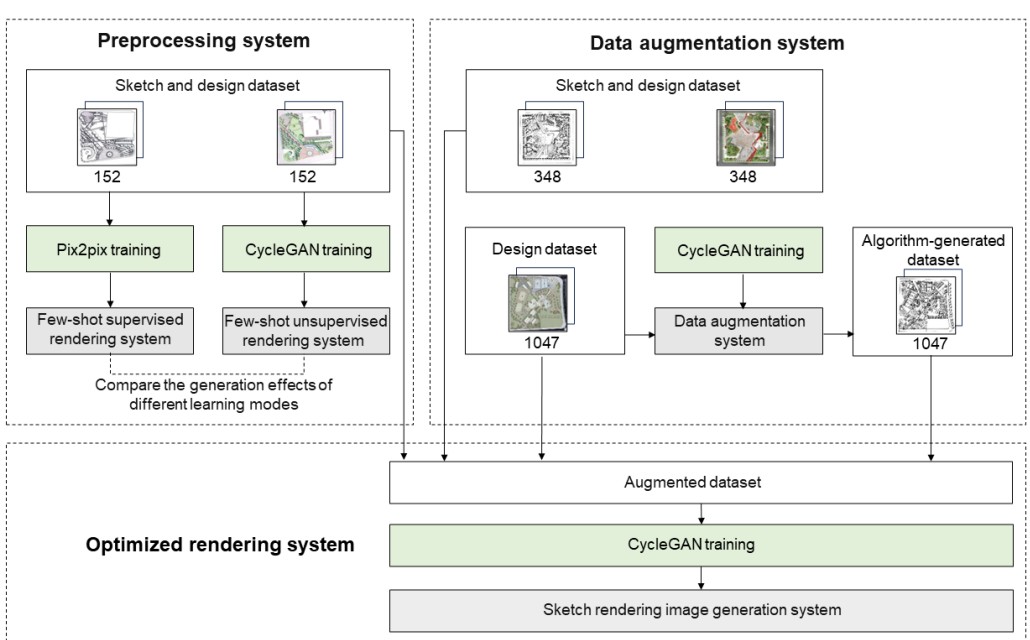

**Figure 5.** This is the framework for the optimized sketch rendering system, including the process from pretraining to data augmentation.

During the pretraining stage, the Pix2pix and CycleGAN algorithms are employed to train the hand-drawn sketch data, each with an equal sample size. The data are processed separately according to the format required by the different algorithms, leading to the initial construction of two small-sample line-draft plan rendering generation systems.

In the data augmentation stage, this study intends to use the CycleGAN algorithm as the foundation. It acquires 348 pairs of line-draft sketches and design drawings from various channels to construct a data augmentation system. Once the data augmentation system is established, we input 1047 design drawings, which lack corresponding line sketches, into the system and obtain 1047 black-and-white line sketches. These algorithmically generated black-and-white line sketches closely resemble human hand-drawn levels in terms of light and dark relationships, line strokes, and other aspects.

Finally, our data are augmented from the original 652 pairs of black-and-white line sketches and design scheme data to 1699 pairs of data samples. By retraining CycleGAN with the augmented dataset, we obtain the optimized design sketch rendering generation system.

*3.5. Testing*

To compare the efficiency of the generation systems built using different algorithms and varying sizes of data samples, this study selected five black-and-white line drafts as the test set. In the testing section, selected samples were inputted into the variously trained stages of CycleGAN and Pix2pix models, yielding post-test images.

From a landscape design perspective, the criteria for selecting test samples primarily included: (1) Park scale sufficient to clearly observe landscape elements such as paving and roads; (2) Rich and diverse vegetation planting methods in the samples; (3) The overall design encompassing a variety of spatial relationships. These were used to test the rendering systems and data enhancement models at each stage, as illustrated in Figure 6.

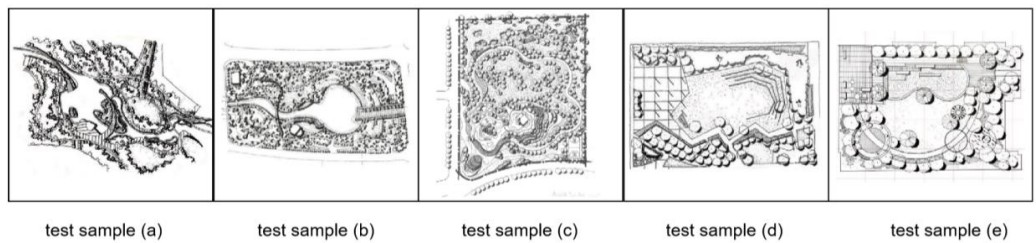

**Figure 6.** These are five representative hand-drawn sketches for testing.

The experimental samples selected are primarily categorized into medium-scale (samples a, b, c) and small-scale park line drafts (samples d, e), each exhibiting distinct characteristics. Planting: samples c and e primarily consist of point-shaped trees, while sample a involves a large cloud tree. The remaining samples comprise a mixed planting of cloud trees and point-shaped trees. Roads and Paving: samples a, b, and c all possess clear main ring roads and branch road structures, exhibiting pronounced spatial opening and closing characteristics. Conversely, samples d and e combine large-area paving and roads. Other Layout Elements: all five samples include lawns of varying areas. Apart from sample d, the rest of the samples contain water bodies.

## 4. Results and Analysis

To evaluate the performance of our developed model, we grouped and compared the test results of five samples. We assessed these results from a landscape design perspective using the following criteria: (1) Whether the color at the edges of the line drawings is clear; (2) Whether details like pavements, roads, and nodes are complete; (3) Whether the colors of trees, lawns, etc., are reasonable and diverse; (4) Whether the overall style is aesthetically pleasing and diverse.

This section primarily analyzes the results of two sets of comparative experiments. The first is a comparison of the small-sample rendering systems of the Pix2pix model and the CycleGAN model. The second is a comparison of the rendering systems of three different data volumes of the CycleGAN model.

*4.1. Algorithm Comparison Evaluation*

In the two small-sample line-draft plane rendering generation experiments, the Pix2pix algorithm and the CycleGAN algorithm exhibited a significant difference in the rendering results, as shown in Figure 7. The experimental results of line-draft rendering indicate that both algorithms have a certain degree of confusion in color expression, and the generated results are not accurate enough. However, the quality of the results generated by the CycleGAN model is significantly superior to that of the Pix2pix model.

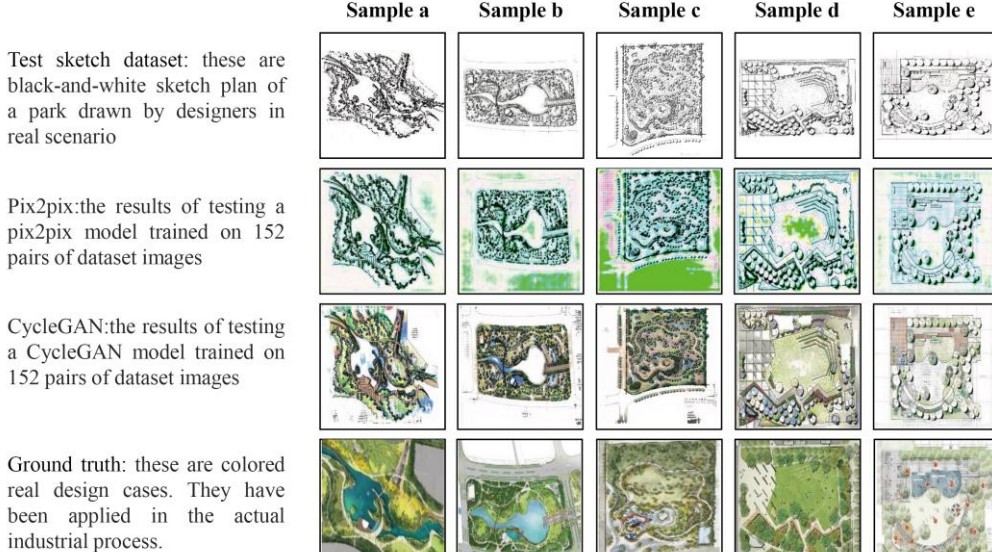

**Figure 7.** This is the comparison of the results generated by Pix2pix and CycleGAN.

In the small-sample test results of CycleGAN, it can be observed that this model can achieve different brightness and changes in the rendering processing of plants (such as samples a, b, c). Simultaneously, it has a certain distinction in the color of paving, water bodies, and lawns (such as samples b, d, e). However, it also has certain shortcomings: (1) It cannot clearly render the color of the water body (such as samples a, b, e). (2) Some trees appear inappropriately blue or purple (such as samples c and d). (3) The boundary color of some roads is blurred (such as samples a, b, c).

In contrast to CycleGAN, the rendering images generated by Pix2pix lack details and real textures, the colors are relatively uneven, and the possibility of blurring is much greater. These problems are because supervised learning needs to use a one-to-one dataset to let the computer learn the transformation logic in it. However, there is a certain gap between the hand-drawn line draft and the design drawing which cannot be completely aligned. Pix2pix makes it difficult to understand the connection between the two, and the results generated are of poor quality.

In the research of style transfer in the field of architectural design, related technologies, such as Pix2pix and CycleGAN, have become mainstream. However, few studies have compared and evaluated the results of the two algorithms in the field. Through the experiments and analysis of this study, the results of CycleGAN in line-draft rendering image generation are more stable and accurate than Pix2pix.

*4.2. Data Volume Expansion Comparison Evaluation*

Although our research indicates that CycleGAN's line-draft rendering is superior to Pix2pix, the generated results still exhibit issues, such as insufficient accuracy and a lack of diversity. Therefore, this study compared the rendering performance of the CycleGAN model under different sample volumes, as depicted in Figure 8. In these test results, models of three different sample sizes reflect differentiated style results.

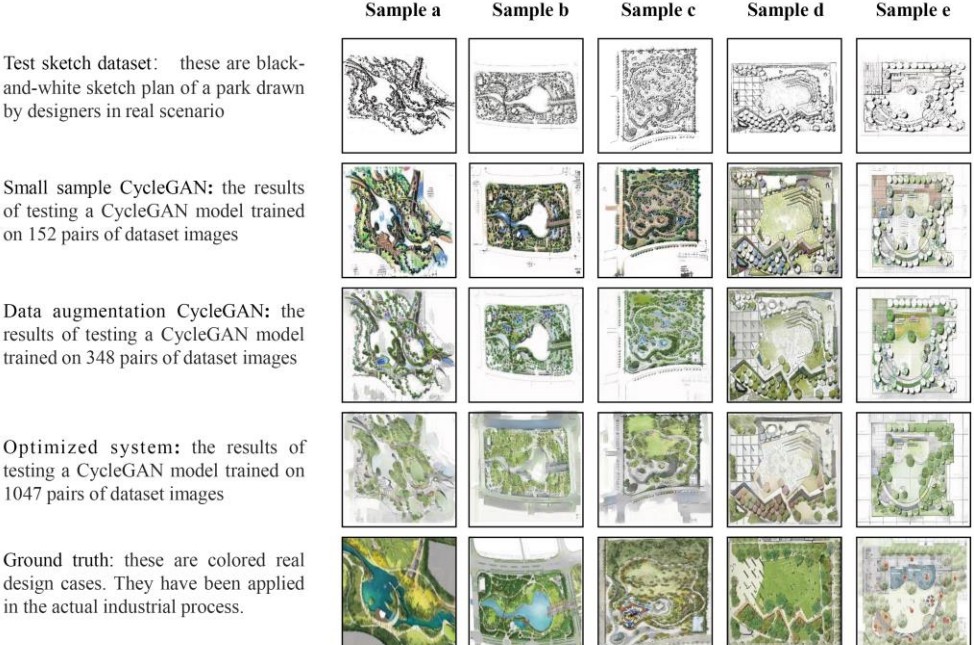

**Figure 8.** This is the comparison of results generated by the model before and after data augmentation.

In the flat rendering experiments of line drafts with two small samples, the Pix2pix algorithm and CycleGAN algorithm demonstrated significant differences in the rendering results. As seen in the comparison from Figure 8, in the landscape sketch rendering models trained with a dataset of 152 pairs, both algorithms exhibited some confusion in color representation and lacked precision in the generated results. However, CycleGAN showed markedly better performance than Pix2pix in terms of color accuracy and richness. Nonetheless, the CycleGAN model still made errors in judging certain detailed elements. The specific details are as follows:

Compared with the small-sample CycleGAN model mentioned earlier, the test results of the data expansion model have the following advantages: (1) It can better distinguish the color of road elements and their boundaries. (2) The expression of tree colors is more accurate, showing different brightness and saturation of green. However, it also has certain disadvantages: (1) Some cloud trees and point-shaped trees show mode collapse in rendering (such as samples a, d, e). (2) It is unable to render the color of water body elements accurately.

Moreover, our optimized system can more accurately and beautifully express rendering results for line drafts of different scales and has a certain style, mainly manifested as: (1) The rendering of the road structure is clear, primarily light gray. (2) The treatment of trees is shown as greens of different brightness, saturation, and transparency, and some lawns have a uniform gradient effect (such as samples a, c, e). (3) There is a certain distinction for the water body, and the rendering effect is significantly higher than the small sample, showing a blue–gray color (such as samples a and b). (4) The overall rendering effect shows a unique style of unified tone.

The comparison in Figure 8 shows that models trained with three different sizes of data samples exhibit varied stylistic results. The small-sample CycleGAN model trained with 152 data pairs showed diverse colors but some areas had colors inconsistent with reality. In contrast, the enhanced CycleGAN model trained with 348 data pairs had a fresher tone and a distinct style. Therefore, we trained an optimized model with 1047 data pairs for comparison. The results indicate that as the training data volume increases, the generated style leans towards a fresher color palette, with more accurate distinctions between roads and lawns, making this style more applicable in practical use. However, the model's ability to distinguish water bodies still needs further enhancement.

The main defects of this model are: (1) The clarity of the water body rendering color is insufficient. (2) The diversity of expressions for paving, nodes, etc., needs to be improved.

Following a comparative analysis, it has been observed that our generation results have significantly improved in several aspects after the data enhancement: (1) For identical plant elements, different brightness, saturation, and green textures can be generated in various positions rather than generating plants of different colors. (2) The interference of noise to the algorithm is relatively reduced, enabling it to accurately identify paving, water bodies, roads, etc., and distinguish color elements. (3) The rendering effect is more accurate and exhibits a certain style.

The experiment demonstrates that this study effectively expands the experimental samples using data enhancement methods, optimizes the model training effect, and generates more realistic and aesthetically pleasing park plan design drawings.

## 5. Discussion

This paper proposes a line-sketch rendering park design system based on Pix2pix and CycleGAN algorithms, which realizes the conversion from black-and-white flat line-sketch drawings to color texture rendering drawings. By utilizing a pretrained model to augment limited sample data, this approach overcomes the constraints of sparse and low-quality data in this field. It involves training an optimized model to enhance the accuracy and variety of generated images, thereby establishing an automated plan rendering system. Additionally, this research has advanced the evolution of human–computer collaborative design workflows within the landscape industry.

In related tasks, such as image coloring and style transfer, the GAN has been widely used and has in-depth research in the coloring of comic sketches and the transformation of different styles of art paintings [36,37]. In the design plan of the construction field, the related research was first implemented in the interior layout design [22]. At the same time, some researchers chose the object of multiple square enclosures composed of space in the Chinese Jiangnan Garden for generation rendering [38]. However, this series of plan rendering research focuses more on the functional area distribution of regular shapes and rarely uses natural shape plan sketches as objects. The field of park green space design, characterized by images rich in complex semantic content, presents a challenge for computers in interpreting the meaning of diverse lines. As a result, a comprehensive research paradigm for this domain has not yet been fully developed.

This study's proposed model enhances the rendering efficiency and can generate images with rich color features. It is capable, to a certain extent, of accurately differentiating various elements in park designs, leading to the production of effective design drawings. Diverging from the traditional approach of coloring based on the image block function and adding details and texture, our model discerns coloring rules from a multitude of hand-drawing design schemes. It extracts the interplay of the texture, shape, color tone, and placement of landscape elements. Consequently, the model can flexibly render plans based on input parameters, enabling the swift generation of results.

Most of the previous related research evaluated the effect of model-generated images from the perspective of computer science. In the research of using line drafts to generate Chinese paintings, Lin and others used the SR method to evaluate the noise of the generated images [39]. In the restoration of Thangka art, Li and others tested the FID distance between the generated image and the actual image and measured the effect of the generation through accuracy [40]. However, this study emphasizes the practicality of the results generated in the related design industry rather than simply conforming to the accuracy of the computer vision field. The results of our model are fresh in color and simple and atmospheric in the picture. This style is expressed as a low-saturation color in computer vision. However, in the actual design workflow, it can better help designers express their thinking and is more suitable for use as actual effect drawings. This study aims to facilitate effective communication between design teams and stakeholders by improving the rendering speed and quality of design drawings. The design process is a collaborative effort requiring

feedback, and by accelerating rendering, we enable designers to present and share ideas more quickly. This speeds up feedback and iteration, enhancing the efficiency of the design process. Faster communication and iteration contribute to better quality and innovation in urban landscape design.

Our research has made progress in rendering design drawings, but it has limitations. The overall rendering effect depends on the quality of the drawings; rough or blurry drawings can lead to distorted results. Our method has yet to adapt to different design styles and aesthetics. Specific limitations include: (1) It is mainly used for small- to medium-scale parks, and may not be accurate for larger landscape designs. (2) It is unable to finely control data postenhancement. Future research could improve in several areas: First, vectorizing hand-drawn sketches to improve the rendering quality and adapt to different design stages and styles. Second, targeting landscape elements in design drawings for specific generation and optimization to enhance the accuracy and intelligence. Third, enhancing model interactivity, integrating with the designer's operational workflow for real-time rendering, quick design adjustments, and improved communication and experience.

While our results indicate that conditional generative adversarial networks are effective in rendering design sketches, this research has certain limitations: (1) The design schemes selected are primarily for small- and medium-sized parks, which may limit the model's ability to generate local nodes in larger-scale park designs accurately. (2) The enhancement of the algorithm does not allow for detailed constraints of the data. In future work, we aim to explore ways to refine the data enhancement methods and improve the algorithm. Our goal is to achieve multiscale plan rendering, thereby enhancing the rendering efficiency and diversifying styles. (3) Our algorithm exhibits limitations in generating vegetation, particularly in the crucial aspect of color diversity. This limitation is not just a technical issue, but also reflects the algorithm's inadequacy in understanding and reproducing the true richness and variety of colors in natural vegetation. Specifically, for vegetation colors other than green, our algorithm fails to capture the subtle nuances and diversity of plant colors found in nature. This issue touches on the profound challenge of bridging algorithmic design with the accurate representation of natural colors, necessitating a better balance between the development of algorithms and a deeper understanding of design theory.

## 6. Conclusions

This paper introduces a method leveraging generative adversarial networks (GANs) designed to rapidly produce rendered drawings from line-drawing plan designs. This method aids designers in swiftly conceptualizing design scenes. The research investigates the influence of the volume and quality of design data, as well as the impact of different algorithms on the generation results driven by algorithmic processes. Additionally, it involves the development of corresponding auxiliary design tools for evaluation.

This paper makes the following two breakthroughs:

Firstly, it addresses the critical challenge of data scarcity in this field. In response, the study introduces a data augmentation model based on generative adversarial networks (GANs) that effectively expands the range of small-sample hand-drawn line-sketch plan drawings.

Secondly, the prevalent color rendering techniques, which primarily depend on edge extraction, risk omitting crucial details in hand-drawn sketches. This loss is particularly evident in the design's structure and texture, significantly hindering designers' capacity to convey their fundamental concepts. To solve these problems, this study has developed a model for rendering directly against hand-drawn line drawings. The model aims to improve work efficiency and generate high-quality color-flat drawings that can be applied in industrial environments.

The experimental results demonstrate the main contributions of this study: (1) The development of a model based on Pix2pix and CycleGAN for rapidly generating diverse and rich design schemes from black-and-white hand-drawn sketches, significantly enhanc-

ing work efficiency. (2) The validation of the scheme's rationality and accuracy through objective evaluation, confirming its applicability in actual design processes and fostering the integration of artificial intelligence technology with landscape design. (3) Empirical evidence from data volume comparison underscores the importance of data augmentation in improving the model quality, leading to the creation of an optimized model with expanded data. (4) The use of hand-drawn sketches as experimental data aligns more closely with the practical application in landscape design processes. This study stands out for its emphasis on practical applications, in contrast to earlier research that mainly followed a computer science research paradigm with a sole focus on image accuracy. It primarily evaluates the generated results from a designer's viewpoint, highlighting its relevance to real-world scenarios. The results generated by the algorithm are expressed as low saturation in color saturation. However, in the actual design workflow, this artistic expression style is more conducive to the expression of designers' thinking and is more suitable for use as actual effect drawings. In the field of landscape design, this study enhances the rendering speed and quality of designs, improving collaboration and communication between designers and clients. While hand-drawn line drafts are indeed the primary embodiment of a designer's conceptualization in the tangible realm, the application of color plays a pivotal role in visual communication. For instance, in the realm of botanical landscaping, while line drafts delineate the hierarchical structure among plants, it is the application of color that critically conveys the harmony of the planting scheme. Similarly, in landscape design illustrations, line drafts establish the relational dynamics among design elements; however, the infusion of color, although not altering the design itself, significantly aids in the accentuation of design areas and focal points, thereby holding substantial value for both the designers and evaluators of multifaceted design proposals.

This research aims to enhance the rendering segment within the intelligent workflow of landscape design. The implementation of more efficient rendering technologies is expected to accelerate the iterative thinking process of designers and enable the vivid presentation of preliminary designs to stakeholders. By reducing the time spent on early design iterations, the overall speed and efficiency of the design process can be significantly improved.

This research initiates an inquiry into the intelligent rendering processes within landscape design. Addressing the challenges encountered in this study, future investigations will delve into specific areas, such as applying constrained color rendering to plant-specific projects and developing renderings in a variety of stylistic floor plans. These topics will direct our forthcoming research efforts, indicating a commitment to advancing the field of landscape design through the integration of innovative rendering techniques. To be more specific, in our future work, we plan to broaden the scope of our design schemes, incorporating the rendering of large-scale urban park line drawings. We will focus on controlling the details of pavements, buildings, and trees to ensure that the local colors in the output results are layered and the textures are richer. Additionally, the model will be fine-tuned to better adapt to various depths in plan sketch inputs, thereby enhancing its efficiency in providing inspiration for designers. We also intend to update related technical methods to achieve superior outcomes. Unlike previous research, this technological process is designed to integrate seamlessly with actual projects, assisting designers in rapidly developing ideas and providing timely feedback to users. This approach is likely to influence the operational modes and personnel composition of some companies. Many related factors arising from this new mode remain to be explored and discussed in future research.

**Author Contributions:** Conceptualization, Writing—Original Draft, R.C.; Funding acquisition, Supervision, J.Z.; Writing—Original Draft, Project administration, X.Y. (Xueqi Yao); Writing—review & editing, Visualization Y.L. and Y.H.; Data curation, Z.L. and H.L.; Methodology, Z.H. and X.Y. (Xingjian Yi). All authors have read and agreed to the published version of the manuscript.

**Funding:** This research was funded by the National Natural Science Foundation of China under Grant 52208041, the Key Laboratory of Ecology and Energy-saving Study of Dense Habitat (Tongji University), Ministry of Education, under Grant 20220110, and the Beijing High-Precision Discipline Project, Discipline of Ecological Environment of Urban and Rural Human Settlements.

**Data Availability Statement:** Dataset available on request from the authors. The raw data supporting the conclusions of this article will be made available by the authors on request.

**Conflicts of Interest:** Haoran Li was employed by the company China United Network Communications Group Co. The remaining authors declare that the research was conducted in the absence of any commercial or financial relationships that could be construed as a potential conflict of interest.

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
