# Peer review of "Enhancing Urban Landscape Design: A GAN-Based Approach for Rapid Color Rendering of Park Sketches"

_land, doi:10.3390/land13020254_

Round 1
Reviewer 1 Report
Comments and Suggestions for Authors
1- There is research for you, that is, for yours. Why did you not refer to it in the sources or references of the studies when you used it?
You are used: (Generative Adversarial Networks (GANs) possess)
Titled:
Chen, R.; Zhao, J.; Yao, X.; Jiang, S.; He, Y.; Bao, B.; Luo, X.; Xu, S.; Wang, C. Generative Design of Outdoor Green Spaces Based on Generative Adversarial Networks. Buildings 2023, 13, 1083. https://doi.org/10.3390/buildings13041083
2- In both pieces of research, you used Conditional Generative Adversarial Networks and (pix).
3- You used Enhancing Urban Landscape Design in The research title.
These words were only mentioned in the title and only appeared once only in the title..
How the research enhances urban Landscape Design, is not clear.
4- On page (9) the standards of landscape design were mentioned.) What are these standards?
5- The two researches are similar in terms of style. The first used pictures and the second used a manual diagram... The research requires more detail than that... Who are the people who drew these diagrams?
Reviewer 2 Report
Comments and Suggestions for Authors
Dear authors, I find your paper interesting. Even though it is scientifically well-structured and presented, it raises the critical question of the proposed tool's purpose for the landscape design process. It is true that this kind of colour rendering could contribute to the efficiency. However, the process of " colouring" is also a part of the landscape design. From the designer's perspective, it is very important that the colour reflects the actual colour of the greenery proposed by the design (for example, what about the Red maple? ) or the actual colour of the pavement, etc. Using the public data on green space design for training is very questionable in terms of the quality of the result. This work represents good exercise from the technological perspective. However, the potential usefulness has to be explained and defined through the perspective of landscape design methodology (missing in the introduction part ) and communication with landscape designers in the first place. It would be interesting to know what professionals think and how they evaluate the results.
Comments on the Quality of English LanguageThere are spelling errors, even in the title.
Reviewer 3 Report
Comments and Suggestions for Authors
The article "Enhancing Urban Landscape Design: A Conditional GAN-Based Approach for Rapid Color Rendering of Park Sketches" explores an innovative system using conditional GANs for transforming black-and-white park sketches into color designs. Please find the suggestion below to improve the article: Introduction: The introduction effectively sets the context. It could benefit from a more detailed explanation of the challenges in current methods, and provides background on the importance of efficient plan rendering in urban landscape design. The literature review is comprehensive but might be improved by directly linking reviewed studies to the specific objectives of the current research. I can suggest to review previous research in image coloring and conditional generative adversarial networks, highlighting the gap in rendering design drawings. Methodology: The methodology section is well-structured, though it could provide more clarity on the choosing specific algorithms. Aslo it is recommende to describes the research framework, including data collection, preparation, and the sketch rendering model using Pix2pix and CycleGAN.
-
- Testing: It is recommende to present the testing methodology and sample selection criteria. Results and Analysis: it is recommende to compare the performance of different models and data volumes in rendering. Also, a deeper critical evaluation of the findings would be beneficial. Discussion: The discussion is insightful; however, it could further explore the implications of the findings in real-world settings, its limitations, and future directions. The conclusion effectively summarizes the research but could emphasize more on the potential impact of the study in the field, and summarize the key findings and contributions of the study.
-
-
Round 2
Reviewer 1 Report
Comments and Suggestions for Authors
Now... good
